# Persistent Lipid Accumulation Leads to Persistent Exacerbation of Endoplasmic Reticulum Stress and Inflammation in Progressive NASH via the IRE1α/TRAF2 Complex

**DOI:** 10.3390/molecules28073185

**Published:** 2023-04-03

**Authors:** Na Lei, Hongfei Song, Ling Zeng, Shaoxiu Ji, Xiangbo Meng, Xiuying Zhu, Xiuyan Li, Quansheng Feng, Jibin Liu, Jie Mu

**Affiliations:** 1School of Basic Medical Sciences, Chengdu University of Traditional Chinese Medicine, Chengdu 611137, China; 2020kb003@stu.cdutcm.edu.cn (N.L.); songhongfei30@163.com (H.S.); jishaoxiu0430b@163.com (S.J.); 2020kb004@stu.cdutcm.edu.cn (X.M.); 2020kb008@stu.cdutcm.edu.cn (X.Z.);; 2School of Clinical Medicine, Chengdu University of Traditional Chinese Medicine, Chengdu 610032, China; 15984022897@163.com

**Keywords:** non-alcoholic steatohepatitis, lipid accumulation, endoplasmic reticulum stress, IRE1α/TRAF2 complex, IKK/IκB/NF-κB signaling pathway, ASK1/JNK1 signaling pathway, inflammation

## Abstract

Non-alcoholic steatohepatitis (NASH) is a metabolic disorder that often leads to other severe liver diseases, yet treatment options are limited. Endoplasmic reticulum (ER) stress is an important pathogenetic mechanism of NASH and plays a key role in tandem steatosis as well as liver inflammation. This study aims to develop a progressive NASH model through sustained lipid accumulation and to elucidate its molecular mechanism through IRE1α/TRAF2 complex. Male SD rats were fed a high-fat diet (HFD) for 4, 8, and 12 weeks to induce progressive NASH. MRNA sequencing and PPI analysis were used to screen core genes. Transmission electron microscopy, immunofluorescence staining, ELISA, qRT-PCR, and Western blotting were used at each time point to compare differences between each index of progressive NASH at 4, 8, and 12 weeks. Sustained lipid accumulation led to structural disruption of the ER, a reduction in ER number, and an increase of lipid droplet aggregation in hepatocytes. Persistent lipid accumulation led to a persistent increase in mRNA and protein expression of the IRE1α/TRAF2 complex, IKK/IκB/NF-κB signaling pathway and ASK1/JNK1 signaling pathway, and TNF-α, IL-1β, and IL-6 also continued to increase. Persistent lipid accumulation led to a persistent exacerbation of ER stress and inflammation in progressive NASH via the IRE1α/TRAF2 complex.

## 1. Introduction

Metabolic-associated fatty liver disease (MAFLD) is the most common chronic liver disease worldwide, which is divided into non-alcoholic fatty liver (NAFL) and non-alcoholic steatohepatitis (NASH) [1,2]. NASH is considered a progressive form of MAFLD and is characterized by hepatic steatosis, inflammation, hepatocellular damage, and varying degrees of fibrosis [3]. NASH refers to a clinical syndrome caused by many factors, including type II diabetes mellitus, hyperlipidemia, and hypertension, but excludes alcohol and other definite liver damage factors. In Europe, the prevalence of NASH is 2% to 5% in the general population and up to 70% in obese individuals [4]. As the obese population increases, the incidence of NASH will continue to rise [5]. NASH increases the risk of developing liver fibrosis, cirrhosis, and even hepatocellular carcinoma (HCC) [6,7], so it is a critical period for clinical intervention.

The endoplasmic reticulum (ER) is an important place for the synthesis, processing, and metabolism of lipids and sterols and is also the most important organelle for maintaining liver lipid homeostasis [8,9]. Lipid, as a stimulatory signal, is prone to interfere with ER function. When the lipid accumulation in the cytoplasm exceeds the metabolic load of the ER, it will initiate a stress mechanism, ER stress. ER stress leads to the activation of the unfolded protein response (UPR), which is mainly mediated by transmembrane protein inositol requiring enzyme 1 alpha (IRE1α), protein kinase R(PKR)-like ER kinase (PERK), and activating transcription factor 6 (ATF6) [10]. In a physiological state, these three transmembrane proteins are bound to B-cell immunoglobulin binding protein/glucose-regulated protein 78 (BIP/GRP78) without exposure [11]. During ER stress, these three transmembrane proteins dissociate from BIP/GRP78, forming three pathways that regulate the quality of folded protein to sense the misfolded protein [12] and reduce protein accumulation. ER stress is connected to multiple biological processes, such as insulin resistance, oxidative stress, mitochondrial dysfunction, and inflammatory response, which makes it a key point in the prevention and treatment of NASH.

During ER stress, the IRE1α pathway is the core pathway that affects lipogenesis and lipid metabolism [10]. Once IRE1α is activated, it recruits TRAF2 to form IRE1α/TRAF2 complex, and this complex interacts with IKK [13], activating IKK/IκB/NF-κB signaling pathway to stimulate a downstream inflammatory response. The activated IRE1α can also activate the TRAF2-ASK1-JNK1/2 complex [14]. Experimental studies have shown that lipid accumulation leads to ER stress [8] and that ER stress is an important factor involved in the development of steatohepatitis and liver fibrosis [15].

As shown through in vitro experiments, previous studies have confirmed that prolonged lipid accumulation leads to ER stress rather than oxidative stress [8]. In this study, blood lipids-related indicators and pathological alterations were analyzed to assess the progressive NASH model. We then used mRNA sequencing and PPI analysis to explore the effect of sustained lipid accumulation on targets and mechanisms involved in the onset of NASH. Furthermore, the relationship between ER stress and downstream signaling pathways was explored by detecting changes in the IRE1α/TRAF2 complex, IKK/IκB/NF-κB signaling pathway, ASK1/JNK1 signaling pathway, and downstream inflammatory factors. This study aimed to provide an experimental and theoretical basis for new targets in the prevention and treatment of NASH.

## 2. Results

### 2.1. Continuous Lipid Accumulation Dysregulated the Concentration of Blood Lipids Related Indicators

As shown in Appendix A (Appendix A), the body weight, visceral fat weight, and body fat percentage of HFD4, HFD8, and HFD12 rats were significantly increased compared to the control groups (*p* < 0.05). This data implicated that unremitting lipid accumulation obviously increased the body weight, visceral fat weight, and body fat percentage in HFD-induced rats.

The concentration of TG, TC, FFA, HDL, and LDL are common indicators to evaluate blood lipids in clinical settings. To explore the effects of lipid accumulation on blood lipids, we tested the concentration of TG, TC, and FFA in liver tissue and HDL and LDL in the serum of rats. The results showed that lipid accumulation obviously increased the concentration of TG, TC, and FFA in liver tissue compared with controls (Appendix A). As shown in Figure 1G,H, sustaining lipid accumulation decreased the concentration of HDL in serum and increased the concentration of LDL in serum. These results demonstrated that continuous lipid accumulation dysregulated the concentration of TG, TC, and FFA in liver tissue and HDL and LDL in serum.

### 2.2. Persistent Lipid Accumulation Caused Hepatocyte Structure Impairment, Collagen Fibers, and Lipid Droplets Accumulation

As shown in Appendix A, compared with control groups, the liver in model groups had a hard texture, earthy yellow color, and blunt edges. To observe the effects of lipid accumulation on the liver structure in rats, we used HE staining to analyze the histopathological changes of the liver. The results showed that persistent lipid accumulation contributed to round vacuoles (balloon-like changes) of different sizes and numbers in the cytoplasm of hepatocytes, with varying degrees of hepatocyte necrosis and inflammatory infiltration (Appendix A). Masson staining was used to identify the collagen fibers in the liver. As per the pictures shown in Appendix A, the blue staining represents collagen fibers. An HFD led to a significant increase in the formation of collagen fibers in the perivascular and confluent areas when compared with controls (*p* < 0.05). The results showed that continuous lipid accumulation led to a significant increase in the percentage of area stained with Masson blue in model groups (*p* < 0.05) (Appendix A). In addition, the relative expression of lipid droplets was calculated by ORO staining. As shown in Appendix A, red staining represents lipid droplets where an HFD significantly increased the percentage area of lipid droplets when compared with control groups (*p* < 0.001). The results indicated that prolonged lipid accumulation resulted in a significant increase in the percentage of ORO staining area and that the percentage of staining area in model groups increased significantly with continuous lipid accumulation (*p* < 0.001) (Appendix A).

### 2.3. Genes Expression Analysis

To further validate the HFD-induced NASH model and to explore the targets and mechanisms of the effects of sustained lipid accumulation on NASH, we performed mRNA sequencing. After sequencing, 38.36~61.88 M clean reads were obtained, and approximately 88.97%~91.83% of the clean reads were localized to the rat reference genome (Appendix A). The sample gene expression and distribution are shown in Appendix A. Next, we also performed sample correlation analysis and principal components analysis (Appendix A).

### 2.4. Differential Expressed Genes (DEGs) Analysis, Protein–Protein Interaction (PPI) Network Construction, and Core Genes Screening

A total of 57,966 DEGs were obtained. Then, 1592 DEGs were screened by the criterion as *p* < 0.05 and |log2Fold Change| > 1. As shown in Appendix A, there was a significant increase in up-regulated DEGs and a significant decrease in down-regulated DEGs with the continuous accumulation of lipids. Next, we analyzed the number of overlap and unique DEGs for each comparison group and showed by Venn plot (Appendix A) that there were 53 overlap DEGs in the three comparison groups.

### 2.5. Function Analysis of Differential Expressed Genes

To elucidate the function of these 53 overlap DEGs, we further performed the Gene Ontology (GO) analysis. GO analysis was divided into three categories, namely biological process (BP), cellular component (CC), and molecular function (MF). We enriched 38 terms, including 30 for BP, 5 for CC, and 3 for MF. The top 10 terms of BP, 5 terms of CC, and 3 terms of MF were displayed in Figure 1A. The BP group focused on biosynthesis and metabolic processes of cholesterol and sterol, as well as liver development; the CC group was the different organelles on which the products of the DEGs were mainly located; and the MF group was the molecular functions possessed by the DEGs.

With the results of GO analysis, we screened out some biological processes related to lipid metabolism and inflammation, such as biosynthesis and metabolic processes of cholesterol and sterol, which validated the HFD-induced NASH model. It was also found that two terms were correlated with ER, and both ranked in the top 5, suggesting that HFD-induced NASH might alter the ER function or structure.

From the results of GO analysis, it was determined that HFD-induced NASH may alter the ER function and structure. The PPI network construction was based on differential genes and ER-associated genes, using the Cytoscape 3.9.1 software for image visualization. As shown in Figure 1B, the network has 109 nodes and 595 edges. The larger and darker the node is, the higher the degree of association between the target protein and other proteins. According to the degree value, the top 15 are TRAF6, IKK-γ, TRAF2, IKK-α, and IKK-β. Due to the large numbers of differential proteins, based on the results of GO analysis and the literature [16,17,18,19,20,21,22,23,24], we selected IRE1α, TRAF2, IKK-β, IκB-α, NF-κB, NF-κB, ASK1, JNK1, TNF-α, IL-1β, and IL-6 as core genes. This was validated by IF, ELISA, qRT-PCR, and Western blotting.

### 2.6. Unremitting Lipid Accumulation Destroys the ER Structure and Increases Numbers of Lipid Droplets

As a result of the GO analysis showing that HFD-induced NASH may lead to the structural changes of the ER, TEM was used to observe the ultrastructural changes of organelles within hepatocytes. As shown in Figure 2A, in the 1000× field of view, the structure of hepatocytes in control groups was intact. There were round and centered nuclei, predominantly euchromatin in the cells, no dilatation of the perinuclear space, and uniform cytoplasmic distribution. In the 5000× field of view, the control groups had abundant ER, which was widely and orderly distributed in the cytoplasm. The ER was not dilated, with a large number of ribosomes attached to the surface. In the 1000× field of view, the structure of hepatocytes in model groups was severely damaged, with swollen and deformed nuclei, irregularly crinkled nuclear membranes, and some nuclei extruded by lipid droplets to the edge of the cells. In the 5000× field of view, the ER of model groups was swollen and ruptured, the number of ER was reduced, and lipid droplets of different sizes were seen to be aggregated and fused in the cytoplasm. The above results indicated that an HFD damaged the ER structure, leading to a decrease in the number of ER and an increase in the number of lipid droplets. The degree of damage to the ER structure and the accumulation of intracellular lipid droplets was more severe as the time of lipid accumulation increased. As shown in Figure 2B, the number of lipid droplets was analyzed in the three HFD groups. The numbers of lipid droplets in HFD12 were significantly higher than in HFD4 and HFD8 (*p* < 0.05), suggesting that the number of intrahepatocellular lipid droplets increases with increasing time of lipid accumulation. Therefore, this is consistent with the results of the GO analysis, as HFD caused an alteration in the ER structure.

### 2.7. Constant Lipid Accumulation Increased the Levels of TNF-α, IL-1β, and IL-6

To further investigate the effect of sustained lipid accumulation on inflammatory factors, we detected the levels of TNF-α, IL-1β, and IL-6 in liver tissue using ELISA. The results showed that an HFD significantly enhanced TNF-α, IL-1β, and IL-6 in liver tissue compared with controls (*p* < 0.05). In the meantime, the levels of inflammatory factors in model groups continued to increase as time prolonged. These findings indicated that ongoing lipid accumulation resulted in increased levels of inflammatory factors (Figure 3A–C).

### 2.8. Sustaining Lipid Accumulation Activated the IRE1α/TRAF2 Complex

To explore the expression of IRE1α and TRAF2, IF staining was conducted to observe and calculate the positive area of IRE1α and TRAF2. IF staining results revealed that lipid accumulation markedly increased the positive expression area of IRE1α and TRAF2 protein compared with controls (*p* < 0.05). The percentage area in model groups increased with the constant lipid accumulation, and the positive expression area of IRE1α and TRAF2 protein in HFD12 was observably higher than HFD4 and HFD8 (*p* < 0.01) (Figure 4A–C).

To deeply investigate the mRNA and protein expression of IRE1α and TRAF2, we performed qRt-PCR and Western blotting. As shown in Figure 4D–H, HFD observably increased the relative levels of IRE1α, TRAF2 mRNA, and protein compared to controls (*p* < 0.05). With the prolonged lipid accumulation, the relative levels of mRNA and protein increased.

### 2.9. Continuous Lipid Accumulation Activated the IKK/IκB/NF-κB Signaling Pathway

To investigate the effect of continuous lipid accumulation on the IKK/IκB/NF-κB signaling pathway, we conducted qRt-PCR and Western blotting to observe the relative expression of mRNA and protein. As shown in Figure 5A–G, HFD dramatically increased the relative levels of IKK-β, IκB-α, NF-κB mRNA, and protein compared with controls (*p* < 0.05). These findings suggested that sustained lipid accumulation triggered an elevation in the relative mRNA and protein expression of the IKK/IκB/NF-κB signaling pathway.

### 2.10. Persistent Lipid Accumulation Activated the ASK1/JNK1 Signaling Pathway

Likewise, we detected the mRNA and protein expression of the ASK1/JNK1 signaling pathway using qRt-PCR and Western blotting. An HFD increased the relative expression of ASK1 and JNK1 mRNA and protein compared to controls (Figure 6A–E). With the persistent lipid accumulation, the relative levels of mRNA and protein increased.

## 3. Discussion

NASH is a serious risk factor for cardiovascular disease, portal hypertension, type II diabetes mellitus, and severe kidney disease. As a result, for these diseases, the risk of adverse outcomes and mortality is heightened [25,26]. NASH is characterized by hepatocellular inflammatory damage and is an important rate-limiting link in the transition from simple fatty liver disease to associated cirrhosis and liver cancer [27,28].

To date, there are no recognized specific drugs for the treatment of NASH [29]. However, there is a great demand for the prevention and treatment of NASH in clinics, highlighting the urgent need to develop effective therapeutic strategies for this disease. Intending to target the pathogenesis and etiology of NASH, several medical treatments with different targets are being tested in clinical trials. Treatments being tested include antioxidants, insulin sensitizers, peroxisome proliferator-activated receptors (PPARs) inhibitors, farnesoid X receptor (FXR) agonists, and intestinal microbiotics. However, these drugs either have limited pharmaceutical efficacy or severe and unpredictable side effects. For example, antioxidant vitamin E has broad specificity, which may adversely affect signaling pathways that require reactive oxygen species [30], and its long-term use may increase the risk of prostate cancer [31]. For insulin sensitizer MSDC-0602K, its improvement of liver histology in NASH is not satisfactory [32]. In dual PPARs inhibitor saroglitazar, its treatment of hepatocellular injury and fibrosis has limited efficacy, and some patients have experienced mild weight gain [33]. Lastly, FXR agonists obeticholic acid and EDP-305 may cause pruritus [34,35].

From the GO analysis results, it appears that HFD-induced NASH may alter the ER function or structure. The ER is an important organelle. It plays a key role in storing calcium, regulating protein translation, and lipid metabolism, as well as having essential relevance to the occurrence of HCC. ER stress is a response that corrects abnormal ER function to maintain ER homeostasis and maintain normal cell function. However, when ER stress is not sufficient to restore ER homeostasis, it will initiate and mediate a variety of biological processes, including the regulation of maturation and secretion of inflammatory factors [13]. In a non-stressed status, IRE1α, PERK, and ATF6 are inactivated by binding to GRP78 [36]. In a stressed status, accumulated unfolded proteins bind to GRP78, causing the above three ER transmembrane proteins to be released from it and become activated [37]. It is known that adipocytes can produce immunomodulatory cytokines such as TNF-α and IL-6 [24]. Lipids also act as stimulatory signals that can directly induce the production of inflammatory cytokines and, as such, induce ER stress to activate downstream inflammatory responses. It has been reported that ER stress is a dominant factor in inflammatory diseases, such as inflammatory bowel disease, type 2 diabetes, and osteoarthritis [38,39,40,41]. Therefore, it is crucial to maintain the homeostasis of all components and mechanisms of the ER [7,11].

Interestingly, from the results of the PPI analysis, IRE1α was screened out. IRE1α is the most prominent and evolutionarily conserved protein among ER membrane proteins [42]. The IRE1α pathway is the central regulator of the protective and injury-inducing mechanisms of the UPR and is the key to initiating biological processes that affect the outcome of stressed cells when ER stress is insufficient to restore ER homeostasis [43]. The IRE1α pathway is also the core pathway by which ER stress triggers UPR to mediate inflammatory responses. The IRE1α/TRAF2 complex is the first signal protein to be activated in the IRE1α pathway, and is the critical signal protein and material basis for the downstream signaling cascade initiated by the IRE1α pathway [10]. Consistent with the above, TRAF2 was also screened out. TRAF2 is essential in the activation of IKK, NF-κB, and JNK, and has also been identified as a key effector molecule in the TNF signaling pathway [16].

Surprisingly, among the top 15 proteins in the ranking degree value from PPI analysis, it was found that several proteins are either core component proteins of the IKK/IκB/NF-κB signaling pathway (IKK-α, IKK-β, IKK-γ, NF-κB) or are involved in the regulation and downstream mechanisms of the NF-κB signaling pathway (TRAF6, TRAF2, TNF). The zinc finger structure of TRAF2 in the IRE1α/TRAF2 complex can activate IKK to release NF-κB dimer and activate the NF-κB signaling pathway and downstream inflammatory responses [44,45]. The transcriptional activation of NF-κB p65/p50 is regulated by IKK, which consists of α and β catalytic subunits and an IKK-γ (NF-κB basic regulator) regulatory subunit [22]. IKK activity leads to the phosphorylation of IκB-α, an inhibitor of NF-κB [22]. Subsequently, NF-κB is released, and it releases a large amount of pro-inflammatory cytokines such as TNF-α and IL-6 [23]. NF-κB plays a vital role in regulating the maturation and secretion of inflammatory factors such as TNF-α, IL-1β, and IL-6 [46,47]. In contrast, TNF-α and IL-1β could promote the activation of NF-κB [24]. Experimental studies have confirmed acute induction of palmitic acid leads to ER stress and triggers IRE1α to form the IRE1α/TRAF2/IKK complex, which promotes NF-κB activation [36]. ER stress is also involved in acute rejection of liver transplantation and lung injury via the IRE1α/TRAF2/NF-κB pathway [48,49].

It is interesting to note that ASK1 and JNK1 were also screened out. One study confirmed that when IRE1α is activated, it recruits TRAF2 to activate NF-κB-mediated pro-inflammatory signaling and JNK-regulated pro-apoptotic signaling [21]. In addition, ER stress activates the ASK1-JNK1/2 signaling pathway, which interacts with ER stress through the IRE1α-TRAF2-ASK1 complex [50]. Numerous reports have shown that during ER stress, IRE1α recruits TRAF2, which then activates ASK1 and, ultimately, the JNK signaling pathway [17,18,19,20]. In conclusion, the IRE1α, TRAF2, IKK-β, IκB-α, NF-κB, ASK1, JNK1, TNF-α, IL-1β, and IL-6 served as the core genes to validate.

TG, TC, FFA, HDL, and LDL are important indicators used to assess the degree of NASH in clinical settings. In this study, we found that prolonged lipid accumulation dysregulated the levels of TG, TC, and FFA in liver tissue and HDL and LDL in serum. Histopathological changes were also used to analyze the degree of hepatocyte damage, liver fibers, and lipid droplet accumulation. HE, Masson, and ORO staining were used to show persistent lipid accumulation increased hepatocellular necrosis, inflammatory infiltration, collagen fibers, and lipid droplet accumulation. MRNA-seq and PPI analysis were used to explore the targets and mechanisms of the effects of sustained lipid accumulation on progressive NASH. Based on the results of mRNA-seq analysis and PPI analysis, we screened out IRE1α, TRAF2, IKK-β, IκB-α, NF-κB, ASK1, JNK1, TNF-α, IL-1β, and IL-6 as core genes to validate. TEM was also used as an essential means of observing the ultrastructure of hepatocytes. Herein, we found unremitting lipid accumulation destroyed the structure of the ER, decreased the number of ER, and increased the accumulation of intracellular lipid droplets. Furthermore, ER stress is one of the most important mechanisms in the occurrence and development of NASH [51,52]. Moreover, another finding of this study was that continuous lipid accumulation increased the mRNA and protein relative expression of IRE1α and TRAF2. In addition, liver inflammation is the hallmark of the clinical diagnosis of NASH, and inflammation plays an important pathological role in NASH, which is closely related to subsequent tissue damage and fibrosis [53,54]. Activated by the IRE1α/TRAF2 complex, the expression of the IKK/IκB/NF-κB signaling pathway and ASK1/JNK1 signaling pathway, and downstream inflammatory factors such as TNF-α, IL-1β, and IL-6 were also aggravated by sustained lipid accumulation.

## 4. Materials and Methods

### 4.1. Animals and High Fat Diet

Several 12-week-old Sprague–Dawley (SD) rats (270 ± 10 g) (Hunan SJA Laboratory Animal Co., Changsha, China; certification no.: SCXK-(XIANG)2019-0004) were raised in common feeding conditions at 22 ± 2 °C, relative humidity of 55%, and a 12 h light/dark cycle for 7 d adaptive feeding.

The HFD consisted of 25% lard, 2% cholesterol, 2% salt, 5% sugar, and 66% common diet, produced by Trophic Animal Feed High-tech Co., Ltd., China (Nantong, China; certification no.: SU (2019)06092). Protocols for experiments involving animals were approved by the Ethics Committee on Laboratory Animals of Chengdu University of Traditional Chinese Medicine. All experiments were strictly in accordance with the guidelines of animal protection, animal welfare, and ethical principles.

### 4.2. NASH Model and Sample Collection

A total of 48 rats were randomly divided into six groups: the control: 4 weeks group (CON4, 8 rats), the high-fat diet: 4 weeks group (HFD4, 8 rats), the control: 8 weeks group (CON8, 8 rats), the high-fat diet: 8 weeks group (HFD8, 8 rats), the control: 12 weeks group (CON12, 8 rats), and the high-fat diet: 12 weeks group (HFD12, 8 rats). CON4, CON8, and CON12 were given common diet and pure water for 4, 8, and 12 weeks, respectively. HFD4, HFD8, and HFD12 were given HFD and pure water for 4, 8, and 12 weeks, respectively. The rats were sacrificed at weeks 4, 8, and 12. The rats were anesthetized, and their blood was obtained through abdominal aortic blood collection. The rats were then euthanized, and their livers were immediately collected for subsequent analysis.

### 4.3. Blood Lipids Related Indicators Analysis

The levels of triglyceride (TG) and total cholesterol (TC) in liver tissue, as well as high-density lipoprotein (HDL) and low-density lipoprotein (LDL) in serum, were detected by an automatic biochemical analyzer (Mindray, Shenzhen, China). The levels of free fatty acid (FFA) in liver tissue were detected by ELISA. TG (105-000449-00), TC (105-000448-00), HDL (105-000463-00), LDL (105-000464-00) kits, as well as automatic biochemical analyzer, were all purchased from Mindray Bio-medical Electronics Co., Ltd. (Mindray, Shenzhen, China). FFA (A042-2-1) kit was purchased from Nanjing Jiancheng Bioengineering Institute (Nanjing Jiancheng, Nanjing, China). All detection processes were performed according to the instructions of kit.

### 4.4. Histological Staining

After being fixed in 4% paraformaldehyde, the liver tissue was wrapped in paraffin and cut into 4 μm slices for (1) hematoxylin–eosin (HE) staining to observe the liver structure and (2) Masson staining to identify the state of collagen deposition. The liver tissue was wrapped in optimal cutting temperature compound embedding agent and cut into 8 μm slices for oil red O (ORO) staining to observe lipid droplets concentration. The Masson and ORO staining results were analyzed by Image-Pro Plus 6.0 software.

### 4.5. MRNA Sequencing

Total RNA was isolated from liver tissue using RNeasy mini kit (Qiagen, 74104, Dusseldorf, Germany). Nano Photometer spectrophotometer (IMPLEN, California, CA, USA) and Agilent 2100 bioanalyzer (Agilent, Palo Alto, California, CA, USA) were used to assess RNA purity and integrity. CDNA syntheses, purification, library constructions, and library quality control were conducted by PMG Co., Ltd. (Guangzhou, China). Finally, the mRNA was sequenced on Illumina NovaSeq 6000 platform (Illumina, California, CA, USA). Raw data were filtered as follows: (1) remove reads with adapters; (2) remove reads with undetermined base information; (3) remove low-quality reads (reads with more than 50% of bases with Qphred ≤ 20). Then, the gained cleaned reads were mapped to the rat genome using Hisat2. Gene abundance was expressed as Fragments Per Kilobase of transcript per Million mapped reads (FPKM). StringTie software was used to count the fragment within each gene, and TMM algorithm of edge R was used for normalization. Differential expression analysis for mRNA was performed using R package DESeq2. The differential expressed genes (DEGs) were screened by criteria as *p* < 0.05 and |log2Fold Change| > 1. GO analysis was performed for biological processes, cellular components, and molecular functions via the David database (https://david.ncifcrf.gov/tools.jsp, 16 November 2022) [55,56].

### 4.6. PPI Network Construction and Gene Screening

Based on the differential genes of interest and ER-associated genes, these were uploaded to the STRING database (https://cn.string-db.org/, 11 March 2023) [57]. The minimum interaction threshold was set as “highest confidence (>0.900)”. The results of the network analysis were imported into Cytoscape 3.9.1 software (NIGMS, Lowell, MA, USA) for results visualization [58].

### 4.7. Transmission Electron Microscopy

For transmission electron microscopy (TEM), liver tissue was fixed in fixative and post-fixed in 1% osmic acid. The liver tissue was then dehydrated with gradient ethanol and acetone penetrated with gradient embedding agent and polymerized. Samples were then cut into 70 nm slices and fixed in 150 mesh cuprum grids with formvar film. Finally, slices were stained with 2% uranium acetate saturated alcohol solution and then stained with 2.6% lead citrate solution. The images were taken by TEM system (Hitachi, Beijing, China).

### 4.8. Enzyme-Linked Immunosorbent Assay

The levels of TNF-α, IL-1β, and IL-6 in liver tissue were detected by rat TNF-α (Elabscience, E-EL-R2856c, Wuhan, China), IL-1β (Elabscience, E-EL-R0012c, Wuhan, China), and IL-6 (Elabscience, E-EL-R0015c, Wuhan, China) ELISA kit. All detection procedures were performed according to the manufacturer’s instructions.

### 4.9. Immunofluorescence Staining

Paraffin-embedded liver sections were deparaffinized with xylene and rehydrated with gradient ethanol. Sections were processed with EDTA antigen retrieval buffer (pH 8.0), 3% hydrogen peroxide (H2O2), and goat serum. Then, sections were incubated with antibodies IRE1α (Proteintech, 27528-1-AP, Wuhan, China), HRP goat anti-rabbit secondary antibodies (Servicebio, GB23303, Wuhan, China), and CY3-TSA (Servicebio, G1223, Wuhan, China). Next, sections were immersed in sodium citrate antigen retrieval buffer (pH 6.0). Additionally, sections were incubated with antibodies TRAF2 (Abcam, ab126758, Cambridge, UK) and secondary antibodies. Then, the nucleus was redyed with DAPI (Servicebio, G1012, Wuhan, China). Finally, sections were processed with spontaneous fluorescence quenching reagent and anti-fade mounting medium. The IF staining results were analyzed by Image-Pro Plus 6.0 software.

### 4.10. Quantitative Real-Time Polymerase Chain Reaction

The total RNA of rat liver tissue was extracted to synthesize the cDNA. The relative mRNA levels of indicated genes were performed with the Taq pro universal SYBR qPCR master mix (Vazyme, Q712-02, Nanjing, China) by qRT-PCR instrument (Analytik Jena, Jena, Germany). Sequences for qRT-PCR primers are presented in Table 1.

### 4.11. Western Blotting

The protein was extracted with RIPA Lysis buffer (Boster, AR0102-100, Wuhan, China) and quantified by ultrastructure nucleic acid protein tester (Analytik Jena, Jena, Germany). Equal amounts of protein were separated by SDS-PAGE, immunoblotted with antibodies IRE1α (Proteintech, 27528-1-AP, Wuhan, China), anti-TRAF2 antibody (Abcam, ab126758, Cambridge, UK), anti-IKK-β antibody (Abcam, ab124957, Cambridge, UK), IκB-α antibody (CST, #4812, Boston, MA, USA), NF-κBp65 antibody (CST, #8242, Boston, US), ASK1 antibody (Proteintech, 67072-1-Ig, Wuhan, China), anti-JNK1 antibody (Abcam, ab199380, Cambridge, UK), and GAPDH antibody (Proteintech, 60004-1-IG, Wuhan, China). Antibody binding was detected using horseradish peroxidase-conjugated anti-rabbit IgG/anti-mouse IgG antibody (Proteintech, SA00001-1/SA00001-2, Wuhan, China) and visualized with Chemiluminescence Solution (Abbkine, BMU102-CN, Wuhan, China).

### 4.12. Statistical Analysis

All data are presented as mean ± SD. Student’s *t*-test was used to compare differences between two groups. One-way analysis of variance (ANOVA) was applied to compare differences between three groups. *p* value less than 0.05 was considered statistically significant. All statistical data were analyzed using SPSS 26.0 (IBM, Chicago, IL, USA).

## 5. Conclusions

In conclusion, these data suggested that continuous lipid accumulation led to increased ER stress and inflammation and triggered a downstream inflammatory response via the IRE1α/TRAF2 complex in progressive NASH. Persistent exacerbation of ER stress and inflammation are important in the pathogenesis of NASH, and our study provides new insights into the pathogenesis of NASH and helps to explore new targets for the prevention and treatment of this disease.

## Figures and Tables

**Figure 1 molecules-28-03185-f001:**
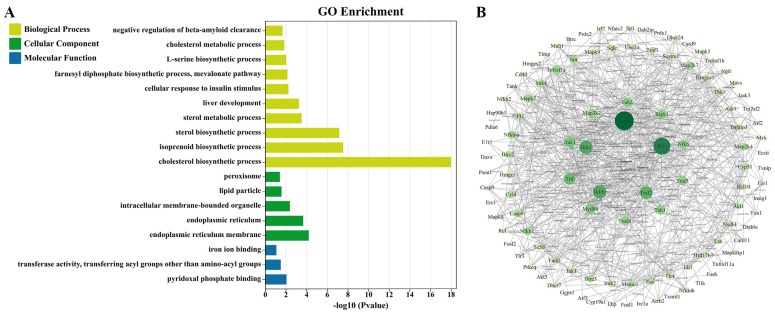
Gene Ontology (GO) analysis of differential expressed genes (DEGs) and diagram of the protein−protein interaction (PPI). (**A**) GO analysis of DEGs. (**B**) Diagram of the PPI: The nodes represent proteins, depending on the degree value, and range in size from small to large and in color from light to dark; The lines represent protein interactions.

**Figure 2 molecules-28-03185-f002:**
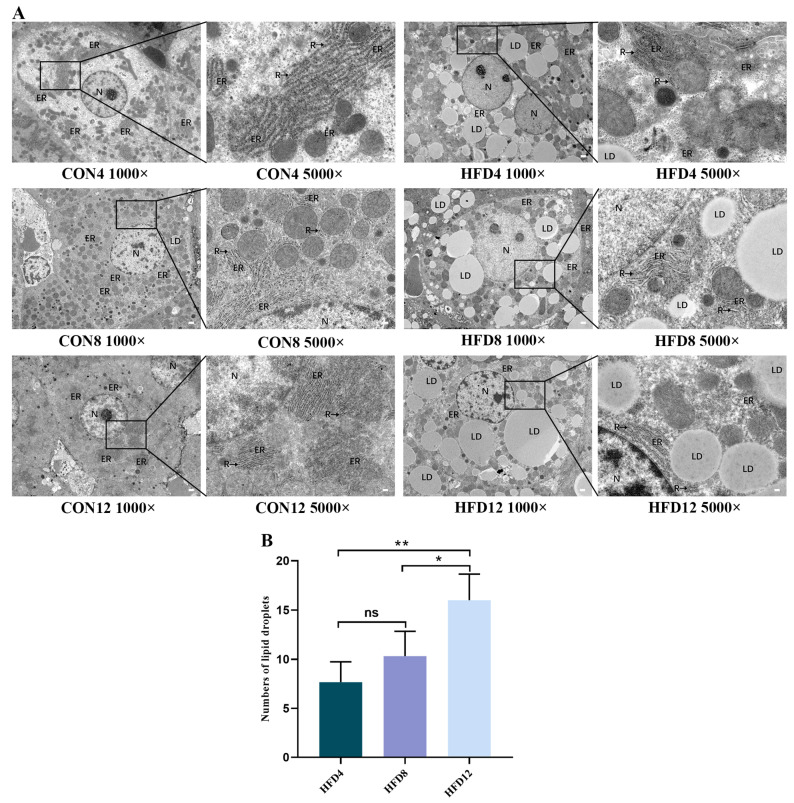
Detection of transmission electron microscopy (TEM). (**A**) The ultrastructure of liver organelles. ER = endoplasmic reticulum, N = nucleus, R = ribosome, LD = lipid droplet. (**B**) The numbers of lipid droplets (*n* = 3, scale bar = 1 µm, diameter of lipid droplets ≥ 3 µm). One-way analysis of variance (ANOVA) was applied to compare differences between three groups. * *p* < 0.05, ** *p* < 0.01. ns = not significant).

**Figure 3 molecules-28-03185-f003:**
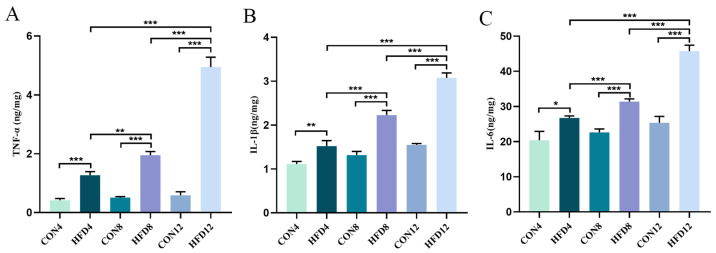
Detection of enzyme-linked immunosorbent assay (ELISA). (**A**–**C**) ELISA of TNF-α, IL-1β and IL-6 (*n* ≥ 4). Student’s *t*-test was used to compare differences between two groups. One-way analysis of variance (ANOVA) was applied to compare differences between three groups. * *p* < 0.05, ** *p* < 0.01, and *** *p* < 0.001.

**Figure 4 molecules-28-03185-f004:**
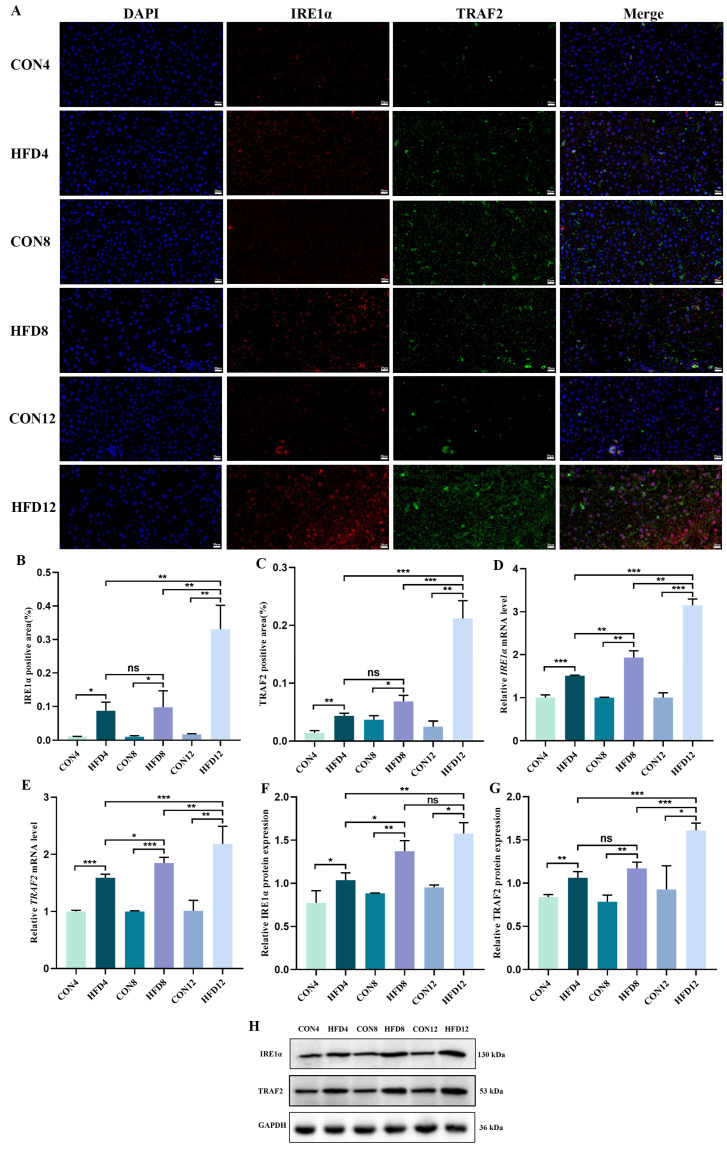
Detection of IRE1α/TRAF2 complex. (**A**) Immunofluorescence (IF) staining (scale bar = 400 µm) of IRE1α and TRAF2. (**B**,**C**) Quantification of IRE1α and TRAF2 (*n* = 3). (**D**,**E**) Relative levels of *IRE1α* and *TRAF2* mRNA (*n* = 3). (**F**–**H**) Relative expression of IRE1α and TRAF2 protein (*n* = 3). Student’s *t*-test was used to compare differences between two groups. One-way analysis of variance (ANOVA) was applied to compare differences between three groups. * *p* < 0.05, ** *p* < 0.01, and *** *p* < 0.001. ns, not significant.

**Figure 5 molecules-28-03185-f005:**
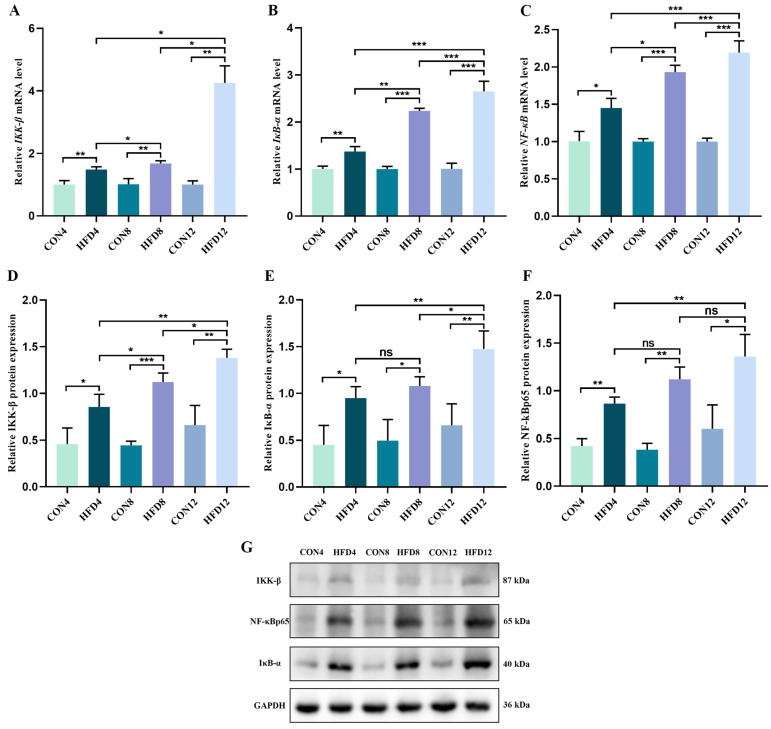
Continuous lipid accumulation activated the IKK/IκB/NF-κB signaling pathway. (**A**–**C**) Relative levels of *IKK-β*, *IκB-α* and *NF-κB* mRNA (*n* = 3). (**D**–**G**) Relative expression of IKK-β, IκB-α and NF-κBp65 protein (*n* = 3). Student’s *t*-test was used to compare differences between two groups. One-way analysis of variance (ANOVA) was applied to compare differences between three groups. * *p* < 0.05, ** *p* < 0.01, and *** *p* < 0.001. ns, not significant.

**Figure 6 molecules-28-03185-f006:**
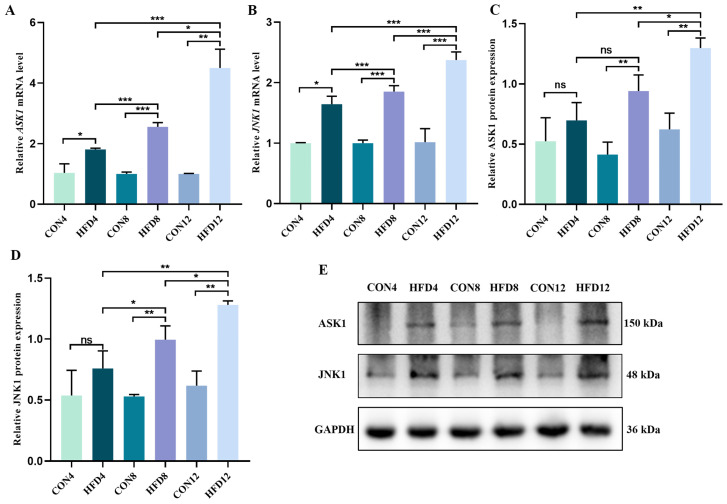
Persistent lipid accumulation activated the ASK1/JNK1 signaling pathway. (**A**,**B**) Relative levels of *ASK1* and *JNK1* mRNA (*n* = 3). (**C**–**E**) Relative expression of ASK1 and JNK1 protein (*n* = 3). Student’s *t*-test was used to compare differences between two groups. One-way analysis of variance (ANOVA) was applied to compare differences between three groups. * *p* < 0.05, ** *p* < 0.01 and *** *p* < 0.001. ns, not significant.

**Table 1 molecules-28-03185-t001:** Primer sequences used in this study.

Gene Name	Forward/Reverse	Primer Sequences (5′-3′)
*GAPDH*	Forward	ACG GCA AGT TCA ACG GCA CAG
	Reverse	GAA GAC GCC AGT AGA CTC CAC GAC
*IRE1α*	Forward	CCT GGC ACT GAA GGT TGG AT
	Reverse	GAG TGG AAG CAG TCA AGG CT
*TRAF2*	Forward	AGC CTT CTT CAC AAG CAG ATA TG
	Reverse	GGT CCA GCA ACA TCA AAG TCA
*IKK-β*	Forward	TGA ACG AGG ATG AGA AGA CTG T
	Reverse	TGG AAG GCT GGG ACA TTA GAT
*IκB-α*	Forward	GTC TCG CTC CTG TTG AAG TG
	Reverse	GTG TCA TAG CTC TCC TCA TCC T
*NF-κB*	Forward	AGA GAA GCA CAG ATA CCA CTA AGA
	Reverse	GTT CAG CCT CAT AGA AGC CAT C
*ASK1*	Forward	ACC TGA ACG CTC CTG GTA CA
	Reverse	TCC TCA GCC AGA AAC CGA CT
*JNK1*	Forward	CTC TCC AGC ACC CGT ACA TC
	Reverse	CGC CAT TCT TAG TTC GCT CC

## Data Availability

The data underlying this article will be shared on reasonable request to the corresponding author.

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
