# Peer review of "Persistent Lipid Accumulation Leads to Persistent Exacerbation of Endoplasmic Reticulum Stress and Inflammation in Progressive NASH via the IRE1α/TRAF2 Complex"

_molecules, 2023, doi:10.3390/molecules28073185_

Round 1
Reviewer 1 Report
The authors describe that in the NASH model developed by high-fat diet feeding, persistent lipid accumulation causes the exacerbation of ER stress and inflammation. Experiments were carried out extensively, and the quality of the work was high. However, the manuscript is merely descriptive and lacks information on molecular mechanisms underlying the phenotypes. Further study is required to address the molecular mechanisms underlying the phenotypes.
Comment 1:
As shown in Fig. 5, the HFD diet significantly increased several inflammatory factors, suggesting that immune cells such as macrophages infiltrate into the liver. It is of interest to see which cells express these factors. The number of Immune cells may correlate with the extent of lipid accumulation and ER stress. Macrophage staining should be compared.
Comment 2:
Since HFD feeding remarkably increases body weight and fat mass, metabolic pathways should be changed in the model rats. Insulin is one of the essential metabolic regulators. Authors should measure insulin levels. Insulin tolerance tests and insulin signaling events are also informative. It is advisable to measure adipokines such as adiponectin and leptin.
Comment 3:
In the discussion part, the authors should describe more mechanistic insights.
Comment 4.
There are English errors throughout the manuscript. I recommend that the whole manuscript is revised by a proofreader or a native speaker of English.
Author Response
Dear Editor,
Thank you for your kind letter regarding manuscript Persistent lipid accumulation leads to persistent exacerbation of endoplasmic reticulum stress and inflammation in progressive NASH via the IRE1α/TRAF2 complex. Based on reviewers’ comments and suggestions, we have made careful modifications to the original manuscript. We hope the revised manuscript will meet the requirements of Molecules. The details of revision is as follows:
Reviewer:
Comments:
(1) As shown in Fig. 5, the HFD diet significantly increased several inflammatory factors, suggesting that immune cells such as macrophages infiltrate into the liver. It is of interest to see which cells express these factors. The number of Immune cells may correlate with the extent of lipid accumulation and ER stress. Macrophage staining should be compared.
Author’s response: Thank you for pointing this out. Macrophage has an important correlation with inflammatory response, which is one of the important factors in the development of NASH. We have performed mRNA-seq with liver tissue, and the mRNA-seq was about the whole liver which contained macrophages. Now we are carrying out mRNA-seq on different single cells in the liver, including macrophages. The aim of this study is to investigate the regulation of HFD-induced lipid accumulation on the overall endoplasmic reticulum (ER) of the liver, therefore no macrophage-related content is presented.
(2) Since HFD feeding remarkably increases body weight and fat mass, metabolic pathways should be changed in the model rats. Insulin is one of the essential metabolic regulators. Authors should measure insulin levels. Insulin tolerance tests and insulin signaling events are also informative. It is advisable to measure adipokines such as adiponectin and leptin.
Author’s response: We appreciate your valuable suggestions. The known pathogenesis of NASH includes “second strike”, genetic susceptibility, insulin resistance, ER stress, mitochondrial dysfunction, dysregulation of intestinal flora and its metabolites, etc. In this study, we found that NASH rats induced by sustained HFD accumulation have correlation with the endoplasmic reticulum and endoplasmic reticulum membrane by result of RNA-seq. Based on the results of RNA-seq and PPI analysis, we investigated the effects of sustained HFD accumulation on ER stress. We present an in-depth explanation of the reasons for choosing ER stress to explore in the Discussion part in Line 290-319, Page 15. However, insulin resistance is one of the important pathogenic mechanisms of NASH. This is an exciting area of investigation for our lab in the future.
(3) In the discussion part, the authors should describe more mechanistic insights.
Author’s response: Thank you for pointing this out. In our revised manuscript, we have added more mechanistic insights in the Discussion part in Line 290-344, Page 15-16. We sincerely hope that you will be satisfied with this manuscript version.
(4) There are English errors throughout the manuscript. I recommend that the whole manuscript is revised by a proofreader or a native speaker of English.
Author’s response: Thank you for pointing this out. We have now worked on language and also have involved native English speaker for language corrections. We really hope that the language level has been substantially improved. The Certificate of English Language Edition has been uploaded as supplementary material.
We are full of gratitude to the reviewers and the editors for their comments and suggestions, which were invaluable in improving the quality of our manuscript. I believe the manuscript has been improved satisfactorily and hope it will be accepted for publication in Molecules. If further modifications are necessary, we will be happy to do so.
Sincerely yours,

Reviewer 2 Report
In this manuscript, Lei et al presented that persistent lipid accumulation can lead to persistent endoplasmic reticulum stress and finally lead to inflammation responses. Although the general finding is interesting, the manuscript is presented in a way that missing coherent logic and the data is very incomplete. Generally, the authors are hitting here and there just to validate what is already published or known to be true. The ER part may be novel, but they only demonstrated a microscopy data to support this finding. I suggest the authors think more about the ER stress part and strengthen the mechanistic study.
Figure 1 and 2 are showing the model is working. This is not new, and a lot of other groups have already demonstrated that. Please consider move these two figures to supplementary figures or merge these two figures.
For figure 1, I suggest the author to use ANOVA based posthoc analysis to avoid multi-comparison problem, although I believe the general conclusion is valid. Same applies to figure5.
For the RNA-seq analysis, first the criterion should be adj.p < 0.05 instead of using P to reduce the number of false positive hits. Second, the differential analysis should not be carried out based on FPKM since these count values are not comparable between samples. The author should use edgeR or DESEQ2 to normalize the count and carry out the downstream analysis.
Next, if the authors aim was to find the DEGs between the HFD and the control group, all three groups should be analyzed together and control for the 4, 8, 12, instead of doing the DE separately and then find the common overlap. This could be easily done in DESeq2 experiment setup, setting experiment design to ~ Treatment+condition. This way, the analysis will output a DEG list from single comparison with the confounding factor controlled.
Following the previous question, the author should perform GO analysis based on the output from the single comparison.
The author failed to report the method they used for GO analysis, but I assume this is based on hypergeometric analysis. Please use GSEA based analysis for a more reliable result.
Line 162 “that DEGs mainly located in the ER…” – the GO results cannot support this statement. In order to make to statement, the authors should provide antibody staining for the proteins. Please be extra careful when making statements.
Figure 4, just by showing representative microscopy figure is not enough. Please find a way to quantify the observation, eg. Mean LD number or size, or ribosome number or size.
The logic to jump to TNF-a, IL1b and IL6 is missing. Did you find enrichment of these genes in your RNA-seq results? Does any inflammation terms get enriched when you do pathway analysis?
It is also hard to follow the logic when talking about IRE1a and TRAF2. There is no background information about this complex, and no explanation supports how the author switch to this complex. Do you also see them in RNA-seq? The authors should really have a coherent logic.
The same problem applies when the authors move to IKK.
Author Response
Dear Editor,
Thank you for your kind letter regarding manuscript Persistent lipid accumulation leads to persistent exacerbation of endoplasmic reticulum stress and inflammation in progressive NASH via the IRE1α/TRAF2 complex. Based on reviewers’ comments and suggestions, we have made careful modifications to the original manuscript. We hope the revised manuscript will meet the requirements of Molecules. The details of revision is as follows:
Reviewer:
Comments:
(1) In this manuscript, Lei et al presented that persistent lipid accumulation can lead to persistent endoplasmic reticulum stress and finally lead to inflammation responses. Although the general finding is interesting, the manuscript is presented in a way that missing coherent logic and the data is very incomplete. Generally, the authors are hitting here and there just to validate what is already published or known to be true. The ER part may be novel, but they only demonstrated a microscopy data to support this finding. I suggest the authors think more about the ER stress part and strengthen the mechanistic study.
Author’s response: Thank you for pointing this out. ER stress, IKK/IκB/NF-κB signaling pathway and ASK1/JNK1 signaling pathway have been shown to be associated with the development of NASH, but our study aimed to focus on the mechanism of the effect of HFD-induced lipid accumulation on NASH, and we linked ER stress to the downstream IKK/IκB/NF-κB signaling pathway and ASK1/JNK1 signaling pathway. There are no reports of HFD-induced lipid accumulation leading to exacerbation of ER stress, exacerbation of the downstream IKK/IκB/NF-κB signaling pathway and ASK1/JNK1 signaling pathway in NASH. In fact, IRE1α is an important membrane protein of the endoplasmic reticulum. When it is activated, it recruits TRAF2 to form the IRE1α/TRAF2 complex, leading to the activation of multiple downstream signaling pathways. In our revised manuscript, we add more contents about ER stress and IRE1α in Line 290-319, Page 15.
(2) Figure 1 and 2 are showing the model is working. This is not new, and a lot of other groups have already demonstrated that. Please consider move these two figures to supplementary figures or merge these two figures.
Author’s response: Thank you for pointing this out. Our focus was to study the differences and progression of HFD-induced lipid accumulation at different time points (4, 8, and 12 weeks), whereas other groups have not yet studied the effect of HFD accumulation on the progression in NASH rats at different time points. Following your suggestion, we have moved Figures 1 and 2 to Figures S1 and S2 in the revised manuscript.
(3) For figure 1, I suggest the author to use ANOVA based posthoc analysis to avoid multi-comparison problem, although I believe the general conclusion is valid. Same applies to figure5.
Author’s response: Thank you for pointing this out. The statistical analysis method you proposed is simple and efficient, but may not be very applicable to our study. Perhaps our statements in the original manuscript were not very clear, which led to misunderstanding, and we sincerely apologize for that. For the statistical analysis of each indicator in Figure 1 and Figure 5, we first compared the model and control groups using student’s t test, and then compared the changes of the indicator in the model groups at 4/8/12 weeks using ANOVA. Our study aimed to explore the targets of HFD-induced lipid accumulation effect. We used the same batch of rats with the same modeling method and different times of sample collection, 4/8/12 weeks, respectively. The control groups were set up to parallel the model groups at the corresponding times, i.e., HFD4 parallel to CON4, HFD8 parallel to CON8, HFD12 parallel to CON12, and HFD4/HFD8/HFD12 also parallel to each other. If all six groups were compared together, the groups would not be parallel to each other. Therefore, we chose the multi-comparison method of student’s t test and ANOVA for statistical analysis. In addition, we have added annotations to the relevant figure legends in Figure legends 1, 2, 3, 4, 5, 6, S1 and S2.
(4) For the RNA-seq analysis, first the criterion should be adj.p < 0.05 instead of using P to reduce the number of false positive hits. Second, the differential analysis should not be carried out based on FPKM since these count values are not comparable between samples. The author should use edge R or DESEQ2 to normalize the count and carry out the downstream analysis.
Author’s response: Thank you for pointing this out. This suggestion you proposed is nice, and using adj.p < 0.05 as criterion might reduce the number of false positive hits. We adopted |log2(FC)| value >1 and p-value <0.05 as screening criteria to keep differentially expressed genes for further analysis. This selection was motivated by the decision to maximize the sensitivity of this analysis for large-scale screening and to identify candidate genes to be validated. In fact, our analysis was done with reference to other experimental studies[1-3], and they all used p<0.05 as a screening criterion in their studies. In addition, we also used |log2(FC)| value >1 and adj.p-value <0.05 as screening criterion for the RNA-seq analysis, and we also enriched some terms in the results of GO analysis, including GO: 0016126~sterol biosynthetic process, GO: 0006695~cholesterol biosynthetic process, and GO: 0005789~endoplasmic reticulum membrane. These are consistent with our previous results. We apologize for the misunderstanding caused by the unclear presentation in our original manuscript. We have added the points you mentioned to the revised manuscript, as follows “Gene abundance was expressed as fragments per kilobase of exon per million reads mapped (FPKM). StringTie software was used to count the fragment within each gene, and TMM algorithm of edge R was used for normalization. Differential expression analysis for mRNA was performed using R package DESeq2.” in Line 422-426, Page 17.
[1] Wang, Z.; Nie, K.; Su, H.; Tang, Y.; Wang, H.; Xu, X.; Dong, H. Berberine improves ovulation and endometrial receptivity in polycystic ovary syndrome. Phytomedicine 2021, 91, 153654, doi:10.1016/j.phymed.2021.153654.
[2] Ye, L.; Wang, L.; Yang, J.; Hu, P.; Zhang, C.; Tong, S.; Liu, Z.; Tian, D. Identification of tumor antigens and immune subtypes in lower grade gliomas for mRNA vaccine development. J Transl Med 2021, 19, 352, doi:10.1186/s12967-021-03014-x.
[3] Liu, C.; Yu, H.; Li, X.; Gong, Y.; Wu, P.; Feng, Q.S. Anti-hepatocellular carcinoma efficacy of Fuzheng Xiaozheng prescription and its interventional mechanism studies. J Ethnopharmacol 2022, 285, 114913, doi:10.1016/j.jep.2021.114913.
(5) Next, if the authors aim was to find the DEGs between the HFD and the control group, all three groups should be analyzed together and control for the 4, 8, 12, instead of doing the DE separately and then find the common overlap. This could be easily done in DESeq2 experiment setup, setting experiment design to ~ Treatment+condition. This way, the analysis will output a DEG list from single comparison with the confounding factor controlled.
Author’s response: Thank you for pointing this out. The analysis method you proposed is common and feasible, but may not be very applicable to our study. Our study aimed to explore the targets of HFD-induced lipid accumulation effect, and we used the same batch of rats with the same modeling method and different times of sample collection, 4/8/12 weeks, respectively. The control groups were set up to parallel the model groups at the corresponding times, i.e., HFD4 parallel to CON4, HFD8 parallel to CON8, HFD12 parallel to CON12, and HFD4/HFD8/HFD12 also parallel to each other. If all six groups were analyzed together, the groups would not be parallel to each other and the screened DEGs might not fit the purpose of our study.
(6) Following the previous question, the author should perform GO analysis based on the output from the single comparison.
Author’s response: Thank you for pointing this out. As answered in the previous question, we think that the method of comparing the three comparison groups separately is more appropriate for this study, so we chose the previous method to perform GO analysis.
(7) The author failed to report the method they used for GO analysis, but I assume this is based on hypergeometric analysis. Please use GSEA based analysis for a more reliable result.
Author’s response: Thank you for pointing this out. We apologize for failing to report the methods we used for GO analysis in the original manuscript. In fact, we used the David database (https://david.ncifcrf.gov/tools.jsp) for GO analysis and have added this method to the revised version, in Line 427-429, Page 17. GSEA is a very advanced analysis method based on whole genome expression set. Thank you very much for this suggestion. Based on your suggestion, we performed the GSEA and the results are shown in the figure below. Consistent with the results of GO analysis of DEGs, GSEA was also enriched to endoplasmic reticulum and endoplasmic reticulum membrane associated terms.

Figure. Gene set enrichment analysis (GSEA) enrichment plot for HFD4 vs CON4, HFD8 vs CON8, and HFD12 vs CON12. (A) Endoplasmic reticulum unfolded protein response. (B) Response to endoplasmic reticulum stress. (C) Endoplasmic reticulum. (D) Nuclear outer membrane endoplasmic reticulum membrane network. (E) Endoplasmic reticulum. (F) Nuclear outer membrane endoplasmic reticulum membrane network. The colored band at the bottom represents the degree of correlation of the expression of these genes (red for a high gene expression and blue for a low gene expression). NES, normalized enrichment score; positive and negative NESs indicate higher and lower expression, respectively.
(8) Line 162 “that DEGs mainly located in the ER…” – the GO results cannot support this statement. In order to make to statement, the authors should provide antibody staining for the proteins. Please be extra careful when making statements.
Author’s response: Thank you for pointing this out. We apologize for the misleading statement in the previous manuscript due to our careless presentation. In our revised manuscript, we have revised this sentence to “HFD-induced NASH might alter the ER function or structure.” in Line 163-164, Page 8.
(9) Figure 4, just by showing representative microscopy figure is not enough. Please find a way to quantify the observation, eg. Mean LD number or size, or ribosome number or size.
Author’s response: Thank you for pointing this out. To quantify the observation of TEM, we added a comparison of the number of lipid droplets in the hepatocytes of the three HFD4/HFD8/HFD12 groups in Line 197-203, Page 8-9, Figure 2B. The details in the manuscript are as follows: As shown in Figure 4B, the numbers of lipid droplets in HFD12 were significantly higher than HFD4 and HFD8 (P < 0.05). It suggested that the number of intrahepatocellular lipid droplets increased with increasing time of lipid accumulation. Consistent with the results of GO analysis, HFD caused an alteration in ER structure.
(10) The logic to jump to TNF-a, IL1b and IL6 is missing. Did you find enrichment of these genes in your RNA-seq results? Does any inflammation terms get enriched when you do pathway analysis?
Author’s response: Thank you for pointing this out. We apologize for the logic missing of TNF-a, IL1b and IL6 by the previous manuscript version. In our newly submitted manuscript, we have supplemented PPI analysis in Line 167-176, Page 8, Figure 1. Interestingly, IRE1α, TRAF2, IKK, NF-kB, ASK1, JNK1 and TNF are all shown in the diagram of PPI analysis. Based on the results of GO analysis and PPI analysis, and literatures, we selected TNF-α, IL-1β and IL-6 as core genes and validated them by ELISA. We sincerely hope that our logic is now easier to follow in this updated version.
(11) It is also hard to follow the logic when talking about IRE1a and TRAF2. There is no background information about this complex, and no explanation supports how the author switch to this complex. Do you also see them in RNA-seq? The authors should really have a coherent logic.
Author’s response: Thank you for pointing this out. We apologize for the incoherent logic in previous manuscript. In our revised manuscript, we have added PPI analysis in Line 167-176, Page 8, Figure 1. Within the results of PPI analysis, we found IRE1α, TRAF2, IKK, NF-kB, ASK1, JNK1 and TNF are all screened out. Based on the results of GO analysis and PPI analysis, and literatures, we selected IRE1α, TRAF2 as core genes and validated them by IF, qRT-PCR and western blotting. Since, once IRE1α is activated, it recruits TRAF2 to form the IRE1α/TRAF2 complex, which then activates the downstream signaling pathway, so we refer to it as a complex. We sincerely hope that our logic is now easier to follow in this updated version.
(12) The same problem applies when the authors move to IKK.
Author’s response: Thank you for pointing this out. We apologize for the poor logic in the previous manuscript. In our revised manuscript, we have supplemented PPI analysis in Line 167-176, Page 8, Figure 1. Within the results of PPI analysis, we found that IRE1α, TRAF2, IKK, NF-kB, ASK1, JNK1 and TNF are all screened out. Based on the results of GO analysis and PPI analysis, and literatures, we selected IKK-β, IκB-α, NF-κB as core genes and validated them by qRT-PCR and western blotting. We earnestly hope that our logic will be easier to understand in this revised manuscript.
We are full of gratitude to the reviewers and the editors for their comments and suggestions, which were invaluable in improving the quality of our manuscript. I believe the manuscript has been improved satisfactorily and hope it will be accepted for publication in Molecules. If further modifications are necessary, we will be happy to do so.
Sincerely yours,

Round 2
Reviewer 1 Report
The authors adequately answered all the questions raised by this reviewer.
Reviewer 2 Report
No more suggestions.